# Machine Learning and Clinical-Radiological Characteristics for the Classification of Prostate Cancer in PI-RADS 3 Lesions

**DOI:** 10.3390/diagnostics12071565

**Published:** 2022-06-28

**Authors:** Michela Gravina, Lorenzo Spirito, Giuseppe Celentano, Marco Capece, Massimiliano Creta, Gianluigi Califano, Claudia Collà Ruvolo, Simone Morra, Massimo Imbriaco, Francesco Di Bello, Antonio Sciuto, Renato Cuocolo, Luigi Napolitano, Roberto La Rocca, Vincenzo Mirone, Carlo Sansone, Nicola Longo

**Affiliations:** 1Department of Electrical Engineering and Information Technology, University of Naples, Federico II, 80100 Naples, Italy; michela.gravina@unina.it (M.G.); carlo.sansone@unina.it (C.S.); 2Department of Neurosciences, Reproductive Sciences and Odontostomatology, University of Naples, Federico II, 80130 Naples, Italy; lorenzospirito@msn.com (L.S.); dr.giuseppecelentano@gmail.com (G.C.); drmarcocapece@gmail.com (M.C.); max.creta@gmail.com (M.C.); gianl.califano2@gmail.com (G.C.); c.collaruvolo@gmail.com (C.C.R.); simonemorra@outlook.com (S.M.); fran.dibello12@gmail.com (F.D.B.); luiginap89@gmail.com (L.N.); mirone@unina.it (V.M.); nicolalongo20@yahoo.it (N.L.); 3Department of Advanced Biomedical Sciences, University of Naples, Federico II, Via S. Pansini, 5, 80131 Naples, Italy; massimo.imbriaco@unina.it; 4Department of Surgery, University of Naples, Federico II, 80130 Naples, Italy; antoniosciuto@gmail.com; 5Department of Medicine, Surgery and Dentistry, University of Salerno, Via Salvador Allende 43, 84081 Baronissi, Italy; renato.cuocolo@gmail.com

**Keywords:** prostate cancer, machine learning, PI-RADS

## Abstract

The Prostate Imaging Reporting and Data System (PI-RADS) classification is based on a scale of values from 1 to 5. The value is assigned according to the probability that a finding is a malignant tumor (prostate carcinoma) and is calculated by evaluating the signal behavior in morphological, diffusion, and post-contrastographic sequences. A PI-RADS score of 3 is recognized as the equivocal likelihood of clinically significant prostate cancer, making its diagnosis very challenging. While PI-RADS values of 4 and 5 make biopsy necessary, it is very hard to establish whether to perform a biopsy or not in patients with a PI-RADS score 3. In recent years, machine learning algorithms have been proposed for a wide range of applications in medical fields, thanks to their ability to extract hidden information and to learn from a set of data without previous specific programming. In this paper, we evaluate machine learning approaches in detecting prostate cancer in patients with PI-RADS score 3 lesions via considering clinical-radiological characteristics. A total of 109 patients were included in this study. We collected data on body mass index (BMI), location of suspicious PI-RADS 3 lesions, serum prostate-specific antigen (PSA) level, prostate volume, PSA density, and histopathology results. The implemented classifiers exploit a patient’s clinical and radiological information to generate a probability of malignancy that could help the physicians in diagnostic decisions, including the need for a biopsy.

## 1. Introduction

Prostate cancer (PCa) is the most frequent male malignancy and the third cause of cancer death in European men [1,2,3,4,5]. Clinical suspicion of PCa is based on an elevated serum prostate-specific antigen (PSA) level and an abnormal digital rectal examination in biopsy-naïve men. However, literature strongly supports the use of multiparametric (mp) MRI before biopsy [6,7], because the latter procedure, if it is not targeted, has low sensitivity and specificity, thus leading to underdiagnosis of clinically significant PCa and to overdiagnosis of non-clinically significant PCa. Indeed, over the last decades, mpMRI has become increasingly valuable for the detection and staging of PCa, gaining a key role in the diagnostic pathway [8]. mpMRI delivers several advantages compared to the systematic transrectal ultrasonography-guided biopsy (TRUSGB) [9]. Firstly, it can rule out non-clinically significant PCa, thus reducing the number of unnecessary prostate biopsies and overdiagnosis. Secondly, it also enables targeted biopsies of suspected lesions [10,11]. Efforts have been made in creating and constantly updating the Prostate Imaging Reporting and Data System (PI-RADS) guidelines that recommend a systematized mpMRI acquisition and define a global standardization of reporting [12]. In particular, the PI-RADS score assigns a numerical value between 1 and 5 to the suspected lesion, correlated with the probability of the lesion being a clinically significant malignancy. However, there is still a lack of consensus on the detailed aspects of mpMRI acquisition protocols and the radiologists’ requirements for reading the examinations [13].

Additionally, the PI-RADS score measures the probability of malignancy and not the PCa aggressiveness. Thus, the biopsy is still needed to assess the clinically significant PCa aggressiveness by measuring the International Society of Urological Pathology (ISUP) Grade Group (GG) and the Gleason Score (GS) [14].

Quantitative assessment of lesion aggressiveness on mpMRI might reinforce the importance, role, and value of MRI in PCa diagnostic, prognostic, and monitoring pathways, providing the radiologist with an objective and non-invasive tool, and thus decreasing intra- and inter-reader variability [15].

Computer-aided design (CAD) and artificial intelligence (AI) are being increasingly explored but require caution. Several studies have shown a limited effect of machine learning (ML)-CAD on prostate MRI reading [16]. In particular, a major issue is that ML-CAD does not achieve stand-alone expert performance [17,18]. ML algorithms are programmed with handcrafted, expert features fed to a simple classifier trained for the diagnostic task. Even though more data has become available, the proficiency of ML-CAD remains below expert performance.

The aim of the paper is to evaluate machine learning (ML) approaches in detecting prostate cancer in patients with PI-RADS score 3 lesions via considering clinical and radiological characteristics. The problem that we are endeavouring to solve can be considered a binary classification task regarding the distinction between patients with and without significant prostate cancer.

The implemented ML models generate as output the probability of malignancy, which could help physicians in diagnostic decisions including the need for a biopsy.

## 2. Materials and Methods

We performed a retrospective data collection from the electronic medical record using a defined source hierarchy. Our dataset, available at the Urologic Unit of AOU Federico II in Naples, consists of 109 patients who underwent trans-rectal prostate biopsy from January to March 2022. All biopsies were performed by the same urologist, with 12 standard plus 2 to 4 target samples in the PIRADS 3 areas detected through fusion-technique. All mpMRI scans were performed and evaluated by a single academic radiologist with extensive expertise in the field. We collected data on patient weight and height, body mass index (BMI), suspect area, prostate volume, prostate-specific antigen (PSA), Psa density, free PSA, ratio, blood glucose, cholesterol, high-density lipoprotein (HDL), low-density lipoprotein (LDL), triglycerides, and creatinine. We collected all data on prostate multiparametric magnetic resonance with indication of PI-RADS v. 2.1 score, and histopathological examinations, performed on the specimen taken during biopsy, provided the PCa aggressiveness by measuring the GS and the ISUP GG, which better reflects PCa biology.

We compared the performance of four machine learning models: classification tree (Ctree), random forest (RF), support vector machines (SVM), and feedforward neural network (NN), which are described below.

Classification tree [19] can be considered a divide and conquer algorithm with recursive iterations. First, an attribute is selected to be placed at the root node, and branches are generated, splitting the instances in subsets. If the attribute can assume a finite set of values, a branch for each of them is generated, while a binary split is computed for numeric attributes. The process can be repeated recursively for each branch, using only the instances that actually reach the branch. If at any time all instances at a node belong to the same class, the process ends for that part of the tree and the node is a leaf node. The predicted class for a new instance is obtained by following the tree from the root down to a leaf node. Since in each node a condition is tested, the classification tree produces a set of IF-THEN rules that can be used for classifying new data.

Random forest [20] is an ensemble learning algorithm that constructs a multitude of classification trees, according to the bagging method. The main idea is that combining the decision of different machine learning models could increase the performance. Random forest takes advantage of the fact that classification trees are very sensitive to data used for the training step by constructing each individual tree with a sample randomly chosen from the dataset with replacement. Moreover, to introduce more variation among the trees, each of them picks only a subset of features.

The idea behind the support vector machines [21] classifiers is to find the boundary between instances belonging to different classes. The algorithm finds the maximum margin hyperplane that is the boundary giving the greatest separation between classes. The instances that are closest to the maximum margin hyperplane are called support vectors, as shown in Figure 1.

However, linear boundaries are not appropriate for all problems. Support vector machines can still be used for nonlinear classification tasks by performing a transformation of variables into a space where the classes are linearly separable. The transformation is performed using kernel functions, as detailed in Table 1.

Feedforward neural network. It is a class of machine learning algorithms inspired by the biological neural network that constitutes animal brains. A neural network is based on a collection of connected units called artificial neurons. The connections between neurons have a weight that increases or decreases the strength of the transmitted information. Precisely, the output of each neuron is computed by multiplying the inputs by the appropriate weights and then summing the results. The sum, plus an extra offset known as the bias, is the input to a function—an activation function—whose output is passed to the next neuron (Figure 2). The weights, the bias, and the activation functions determine how the inputs are transformed into outputs.

In neural networks, neurons are organized in layers:-The input layer receives the input variables.-The hidden layer is the collection of neurons with activation functions. It is the layer responsible for the extraction of the features from the input data.-The output layer produces the result for given inputs.

In a feedforward neural network, information is passed or fed forward from one layer to the next. Each neuron is connected to every neuron in the previous layer.

For pre-processing, the features weight and height were excluded due to their correlation with body mass index (BMI). For each patient, we added a new feature representing the number of suspected areas (TOT_ZONE). The feature suspect area was encoded as a vector where the i-th element is set to 1 if the corresponding area was suspected. The result is a dataset with all numeric features.

The dataset was normalized using z-score normalization, and we used adaptive synthetic sampling (Adasyn) [22] to handle the high imbalance between patients with and without malignant lesions. This method creates synthetic samples to balance the minority class. More specifically, it finds the k-nearest neighbours of each minority example and calculates a value that indicates the dominance of the majority class in each specific neighbourhood. Then, it generates synthetic data for each neighbourhood.

For features selection, we searched for discriminative features via implementing a features selection step. In particular, we used backward features elimination, which is a wrapper approach that is able to discover feature dependencies by taking into account the selected machine learning model. Backward elimination is an iterative process: in the beginning, all the features are considered, and at each iteration the algorithm removes the least significant feature which improves the performance of the model.

For model training, we compared the performance of the different machine learning algorithms: classification tree (Ctree), random forest (RF), support vector machines (SVM) and neural network (NN). After a model optimization step, in the classification tree model, the minimum number of leaf node observations was set to 4, and the split criterion was the Gini’s diversity index. The number of trees in random forest was set to 273, while the support vector machines algorithm used a linear kernel. The implemented neural network consisted of two fully connected layers, followed by rectified linear unit (ReLU) activation function (Figure 3).

The experiments were performed using a 10-fold cross validation, and performance were evaluated in terms of accuracy (ACC), specificity (SPE), sensitivity (SENS), F1-score (F1), and area under the ROC curve (AUC). The described approach was implemented in MATLAB 2020b.

## 3. Results

Data on clinical characteristics from 109 consecutive patients who underwent mpMRI and transrectal prostate biopsy are reported in Table 1. The median age reported was 67 (58–79) years old, while median PSA was generally over the cut-off for prostate biopsy.

All patients received a PI-RADS V2.1 score of 3, a histopathological diagnosis of prostate cancer with Gleason Score (reported in Table 1), and ISUP risk classification score or absence of prostate cancer. Fifty patients had no tumour, whereas PCa was reported for 59 patients.

Table 2 reports the results of the implemented approach, while Table 3 shows the features selected by each model with the features selection step.

Random forest (RF) showed the best performances, reporting an AUC of 83.32%, and outperforming all the other models in accuracy, sensitivity, and F1-score. Moreover, the high sensitivity (81.69%) suggests that the model is able to recognize patients belonging to the malignant class, the most critical class.

All models in the study showed validity in predicting the need for biopsy.

## 4. Discussion

This study aimed to predict PCa aggressiveness using ML techniques on quantitative mpMRI data. In particular, we focused on peripheral lesions considered radiologically indeterminate (with PI-RADS = 3) and examined according to PI-RADS 2.1 guidelines.

The most important claim of prostate MRI is that it can avoid unnecessary biopsies, but to optimally achieve this goal requires expert performance, with high negative predictive value, and good image quality. Experts specifically mention these as requirements

We combined mpMRI data with clinical data exploring the power of prediction of four ML models, namely random forest, classification tree, neural network, and support vector machines. More specifically, we analysed the performance of the ML models in PCa aggressiveness prediction via considering patients’ clinical data with the aim of providing physicians with a decision support system. Since the algorithms are very sensitive to the involved features, we also implemented a feature selection step in order to determine the most important clinical characteristics for each model, as reported in Table 3.

In a previous study, the detection rate of PCa in MRI fusion biopsy of PI-RADS 3 lesions alone ranged from 16% to 35% [7,23,24], which is significantly inferior to our detection rate with ML models of 71% to 83.3%.

In previous studies, researchers used new measures. Hansen et al. [25] studied the different locations of PI-RADS 3 lesions; the detection rate of PI-RADS 3 PCa was 21%. For peripheral lesions, the CDRs differed according to the round shape of lesions (*p* = 0.0055) and ADC value (*p* = 0.0001). For transitional lesions, high CDR was associated with a more anterior location (*p* = 0.0048), a more ill-defined boundary (*p* = 0.0092), and a lower ADC value (*p* = 0.0057). However, a recent study showed no significant difference in median ADC values on univariate analysis (*p* = 0.112) [26]. In our study, we did not develop new measurement indicators but rather used radiological and clinical data combined with machine learning–based algorithms, one of the important branches of AI, which has been developing rapidly in recent years and has been applied in biometric recognition, medical diagnosis, etc.

As reported in Table 2, random forest showed the best performance. In our best prediction model, the sensitivity reached 81.69% with the specificity of 71.05%, resulting in a good ability to recognize the malignant class. Although the SVM and the Ctree models showed the highest sensitivity (73.68%), we chose RF as the best model as it outperformed the other models in all other metrics, whilst maintaining a good value for specificity (−2.63%).

The implemented model could be easily used for the PI-RADS score 3 both for individual patients and for lesions. It is able to effectively use a patient’s clinical information in order to quickly indicate PCa aggressiveness.

Moreover, the probability of malignancy suggested by the implemented model can be useful for estimating an order of severity among different patients, determining not only the need for biopsy, but also its urgency.

To minimize unnecessary biopsies with minimal missed diagnoses, clinicians could use the prediction of the classifier as a reference for clinical decisions that would be most beneficial to patients with PI-RADS 3 lesions. However, since our study was retrospective, prospective validation is still needed.

The proposed methodology shows very promising results (Table 2), confirming the applicability of ML approaches in systems supporting physicians in diagnostic decisions.

However, our study has some limitations. First, it was retrospective, and the amount of data involved in the study was small. Second, the differences in US diagnostic hardware and software used in the fusion process might also have caused some bias. Third, the study is monocentric. Adding more information (DCE, familiarity, etc.) to the ML model would be likely to provide further improvements; moreover, an external validation cohort should be included to test the reproducibility of the established method in future.

A larger dataset leads to improved performance, which can potentially reach expert-level performance when substantially more than 2000 training cases are used.

## 5. Conclusions

In this study, a machine-aided system was developed to detect clinically relevant PCa. This machine learning approach has the potential to improve the performance of a structured PIRADS v 2.1 scheme by providing radiologists and urologists with quantitative and standardized criteria, thereby enabling them to more confidentially detect cancer for better patient counselling and treatment planning. Further studies are needed to better implement machine learning approaches and AI technology.

## Figures and Tables

**Figure 1 diagnostics-12-01565-f001:**
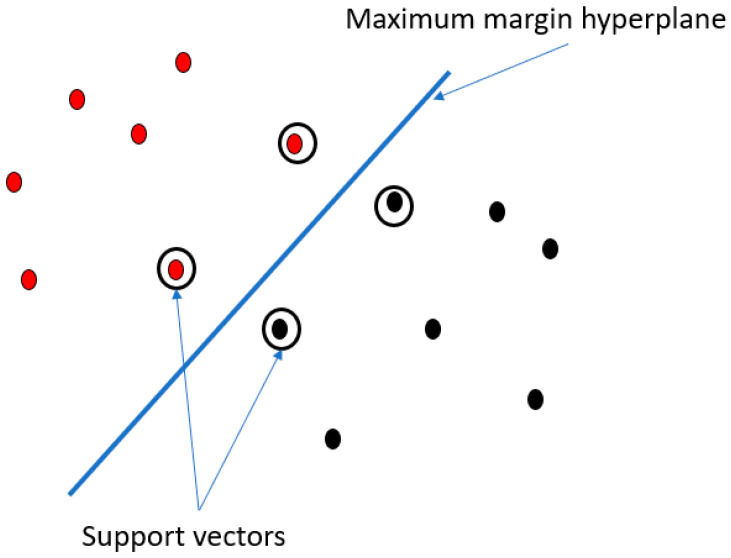
The figure shows the maximum margin hyperplane and support vectors. Points with different colours represent instances of different classes (the red and the black one).

**Figure 2 diagnostics-12-01565-f002:**
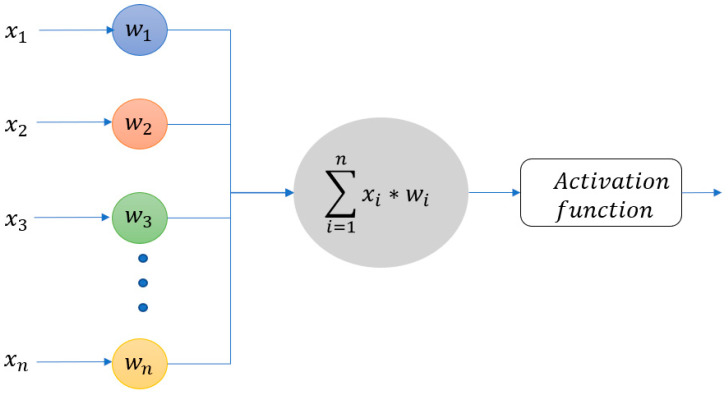
The figure shows how each neuron computes the output. The vector x = (x_1_, x_2_, x_3_, …, x_n_) is the input, while the vector w = (w_1_, w_2_, w_3_, …, w_n_) represents the weight for each connection.

**Figure 3 diagnostics-12-01565-f003:**
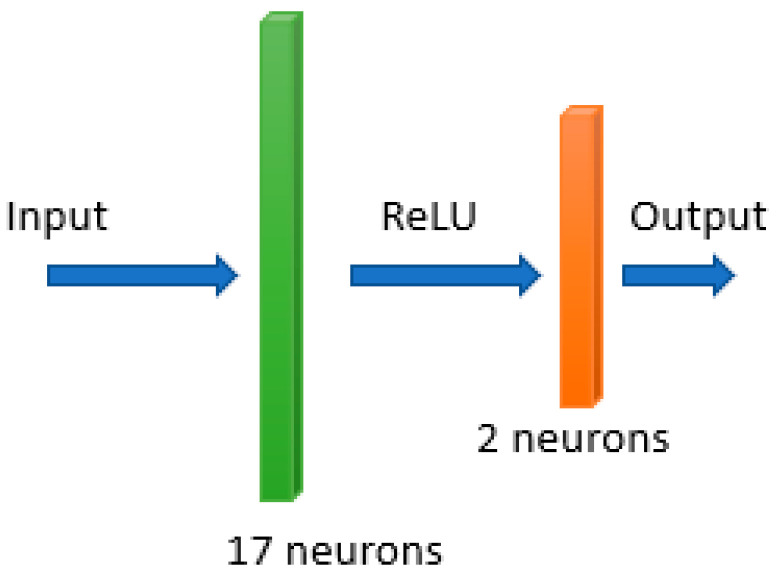
Architecture of the implemented neural network.

**Table 1 diagnostics-12-01565-t001:** General characteristics of the study population.

Age (years)	**Median**	67
	**IQR**	58–79
BMI	**Median**	26.8
	**IQR**	18.2–34.9
Prostate volume, gr.	**Median**	48
	**IQR**	19–138
PSA, ng/mL	**Median**	6.2
	**IQR**	0.24–15.43
PSA density	**Median**	0.13
	**IQR**	0.01–0.8
Serum Glucose, mg/dL	**Median**	95
	**IQR**	73–196
Serum Creatinine, mg/dL	**Median**	1.03
	**IQR**	0.79–1.84
Gleason Score 6 (3 + 3)	**N. of patients**	18
Gleason Score 7 (3 + 4)	**N. of patients**	25
Gleason Score 7 (4 + 3)	**N. of patients**	17
Gleason Score 8 (4 + 4)	**N. of patients**	6
Gleason Score 9 (4 + 5)	**N. of patients**	3

BMI: Body mass index; IQR: interquartile range; PSA: prostate-specific antigen.

**Table 2 diagnostics-12-01565-t002:** Results of the implemented experiments in 10-fold cross-validation.

Method	ACC	SPE	SENS	F1	AUC
RF	77.98%	71.05%	81.69%	82.86%	83.32%
NN	70.53%	53.33%	78.46%	78.46%	74.51%
Ctree	74.31%	73.68%	74.65%	79.10%	74.30%
SVM	72.48%	73.68%	71.83%	77.27%	72.76%

ACC: accuracy; AUC: area under the ROC curve; Ctree: classification tree; F1: F1-score; NN: neural network; RF: random forest; SENS: sensitivity; SPE: specificity; SVM: support vector machines.

**Table 3 diagnostics-12-01565-t003:** For each machine learning algorithm, the selected features are reported.

Method	Selected Features
RF	BMI-equator-apex-TOT_ZONE-PSA density-ratio-Blood glucose-HDL-Triglycerides-Creatinine -
Ctree	TOT_ZONE-prostate volume-Blood glucose-HDL-Triglycerides-
NN	BMI-base-equator-apex-transitional-TOT_ZONE-prostate volume-PSA-psa density-Free PSA-ratio-Blood glucose-Total Cholesterol-HDL–LDL-Triglycerides-Creatinine-
SVM	BMI-base-TOT_ZONE-PSA-psa density-ratio-Blood glucose-Triglycerides-Creatinine-

BMI: body mass index; Ctree: classification tree; HDL: high-density lipoprotein; LDL: low-density lipoprotein; NN: neural network; PSA: prostate-specific antigen; RF: random forest; SVM: support vector machines; TOT_ZONE: number of suspected areas.

## Data Availability

Not applicable.

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
