# Peer review of "Machine Learning and Clinical-Radiological Characteristics for the Classification of Prostate Cancer in PI-RADS 3 Lesions"

_diagnostics, 2022, doi:10.3390/diagnostics12071565_

Round 1

Reviewer 1 Report

In the submitted manuscript entitled "Machine Learning and clinical-radiological characteristics for the classification of prostate cancer in PI-RADS 3 Lesions", the authors studied the ability to detect prostate cancer in patients with PI-RADS 3 lesions using 4 machine learning models. The results are well interpreted and explained. 
The manuscript in its current form is suitable for publication with minor revision mainly with regard to grammar and language consistency. Also, the conclusion section needs to be revised to reflect the results of the study.

Author Response

We thank the Reviewer for the comments and suggestions. We improved the quality of the manuscript by performing an extensive grammar review. Moreover, as suggested by the Reviewer, we revised and improved the conclusion section as which now reads as follow:

(p. 8, line 253-258): “In this study, a machine-aided system was developed to detect clinically relevant PCa. This machine learning approach has the potential to improve the performance of a structured PIRADS v2.1 scheme by providing radiologists and urologists with quantitative and standardized criteria, thereby enabling us to more confidentially detect cancer for better patient counseling and treatment planning. Further studies are needed to better implement machine learning approaches and AI technology.”

Reviewer 2 Report

I read with interest this nicely written manuscript. The field and the subject is of sure interest.

However, I have some minor issues that I believe should be addressed:

-        -Transrectal prostate biopsies were fusion biopsies?

-       - Which was the template? 12 standard samples? Or with the addition of targeted samples in the PIRADS 3 areas (cognitive/fusion?). This is not completely clear. It looks like the biopsies were not fusion. In this case it should be declared in limitations section at the end of the discussion. Please also better specify the type of biopsy in the methods.

-        -Were the prostate biopsies performed by the same urologist / urologists with similar experience in PBX?

-       - Please explain the abbreviation in the tables: NN, Ctree, etc…

-      -  In table 1: I don’t understand the p-value. In my opinion, this is a descriptive stat, it doesn’t need a p-value unless it is about the difference between the two samples: with PCa and without PCa. However, this is not clear in the methods, nor in the table legend.

-        -“In a previous study, the detection rate of PCa in MRI fusion biopsy of PI-RADS 3 lesions alone ranged from 16 to 35% [18-20], which is significantly inferior to our detection 195 rate with ML models of 71% - 83.3%”. This is true. However, the detection rate in your sample, without the use of ML is more than 50% (59 out of 109). This was probably a favorable sample, that is fine, but you should at least declare it in the discussion since the “help” provided by the ML make a strong contribution but from a 50% to a 80% detection rate (not from a 20% to a 80%).

Author Response

  • Point 1: Transrectal prostate biopsies were fusion biopsies? Which was the template? 12 standard samples? Or with the addition of targeted samples in the PIRADS 3 areas (cognitive/fusion?). This is not completely clear. It looks like the biopsies were not fusion. In this case it should be declared in limitations section at the end of the discussion. Please also better specify the type of biopsy in the methods.

Response 1: We thank the Reviewer for the comments and suggestions. We revised and improved the Materials and Methods section as which now reads as follow:

(p. 2, line: 79-81): “All biopsies were performed by the same urologist, with 12 standard plus 2 to 4 target samples in the PIRADS 3 areas detected through fusion-technique.”

  • Point 2 : Were the prostate biopsies performed by the same urologist / urologists with similar experience in PBX?

Response 2: We thank the Reviewer for the comments and suggestions. We revised and improved the Materials and Methods section as which now reads as follow:

(p. 2, line: 79-81): “All biopsies were performed by the same urologist, with 12 standard plus 2 to 4 target samples in the PIRADS 3 areas detected through fusion-technique.”

  • Point 3: Please explain the abbreviation in the tables: NN, Ctree, etc…

Response 3: We thank the Reviewer for the comments and suggestions. We added abbreviations at the bottom of the tables which now look like this:

Table 1. Study population, general characteristics.

Age

(years)

Median

67

IQR

58-79

BMI

Median

26,8

IQR

18,2 - 34,9

Prostate volume, gr.

Median

48

IQR

19-138

PSA, ng/ml

Median

6,2

IQR

0,24-15,43

PSA density

Median

0,13

IQR

0,01-0,8

Serum Glucose, mg/dl

Median

95

IQR

73-196

Serum Creatinine, mg/dl

Median

1,03

IQR

 0,79-1,84

Gleason Score 6 (3+3)

N. of patients

18

Gleason Score 7 (3+4)

N. of patients

25

Gleason Score 7 (4+3)

N. of patients

17

Gleason Score 8 (4+4)

N. of patients

6

Gleason Score 9 (4+5)

N. of patients

3

BMI: Body mass index; IQR: interquartile range; PSA: prostate-specific antigen.

Table 2. Results of the implemented experiments in 10-fold cross-validation.

Method

ACC

SPE

SENS

F1

AUC

RF

77,98%

71,05%

81,69%

82,86%

83,32%

NN

70,53%

53,33%

78,46%

78,46%

74,51%

Ctree

74,31%

73,68%

74,65%

79,10%

74,30%

SVM

72,48%

73,68%

71,83%

77,27%

72,76%

ACC: Accuracy; AUC: Area under the ROC Curve; Ctree: Classification Tree; F1: F1-Score; NN: Neural Network; RF: Random Forest; SENS: Sensitivity; SPE: Specificity; SVM: Support Vector Machines.

Table 3. For each Machine learning algorithm, the selected features are reported.

Method

Selected Features

RF

BMI -    equator -    apex -    TOT_ZONE -    psa density -    ratio -    Blood glucose -    HDL -    Triglycerides -  Creatinine -

Ctree

TOT_ZONE -    prostate volume -   Blood glucose -    HDL -    Triglycerides -

NN

BMI -    base -    equator -    apex -    transitional -    TOT_ZONE -    prostate volume -    PSA -    psa density -  Free PSA -    ratio -    Blood glucose -  Total  Cholesterol -    HDL – LDL -    Triglycerides -    Creatinine -

SVM

BMI -    base -    TOT_ZONE -    PSA -    psa density -    ratio -   Blood glucose -    Triglycerides -    Creatinine -

BMI: Body mass index; Ctree: Classification Tree; HDL: High-density lipoprotein; LDL: Low-density lipoprotein; NN: Neural Network; PSA: prostate-specific antigen; RF: Random Forest; SVM: Support Vector Machines; TOT_ZONE: number of suspected areas.

  • Point 4:  In table 1: I don’t understand the p-value. In my opinion, this is a descriptive stat, it doesn’t need a p-value unless it is about the difference between the two samples: with PCa and without PCa. However, this is not clear in the methods, nor in the table legend.

Response 4: We thank the Reviewer for the comments and suggestions. The column reporting the p-value represents an error in the transposition of the manuscript in the journal template. We corrected this error by eliminating the column, which now looks like this:

Age

(years)

Median

67

IQR

58-79

BMI

Median

26,8

IQR

18,2 - 34,9

Prostate volume, gr.

Median

48

IQR

19-138

PSA, ng/ml

Median

6,2

IQR

0,24-15,43

PSA density

Median

0,13

IQR

0,01-0,8

Serum Glucose, mg/dl

Median

95

IQR

73-196

Serum Creatinine, mg/dl

Median

1,03

IQR

 0,79-1,84

Gleason Score 6 (3+3)

N. of patients

18

Gleason Score 7 (3+4)

N. of patients

25

Gleason Score 7 (4+3)

N. of patients

17

Gleason Score 8 (4+4)

N. of patients

6

Gleason Score 9 (4+5)

N. of patients

3

Table 1. Study population, general characteristics.

BMI: Body mass index; IQR: interquartile range; PSA: prostate-specific antigen.

  • Point 5: “In a previous study, the detection rate of PCa in MRI fusion biopsy of PI-RADS 3 lesions alone ranged from 16 to 35% [18-20], which is significantly inferior to our detection rate with ML models of 71% - 83.3%”. This is true. However, the detection rate in your sample, without the use of ML is more than 50% (59 out of 109). This was probably a favorable sample, that is fine, but you should at least declare it in the discussion since the “help” provided by the ML make a strong contribution but from a 50% to a 80% detection rate (not from a 20% to a 80%).

Response 5: We thank the Reviewer for the comments. Our analyses were performed by a single academic radiologist with very high experience in prostatic MRI evaluation as which now reads as follow:

(p. 3, line: 81-82): “All mpMRI were performed and evaluated by a single academic radiologist with extensive expertise in the field.”
